# Patients with Proliferative Lupus Nephritis Have Autoantibodies That React to Moesin and Demonstrate Increased Glomerular Moesin Expression

**DOI:** 10.3390/jcm10040793

**Published:** 2021-02-16

**Authors:** Dawn J. Caster, Erik A. Korte, Michael L. Merchant, Jon B. Klein, Michelle T. Barati, Ami Joglekar, Daniel W. Wilkey, Susan Coventry, Jessica Hata, Brad H. Rovin, John B. Harley, Bahram Namjou-Khales, Kenneth R. McLeish, David W. Powell

**Affiliations:** 1Department of Medicine, University of Louisville School of Medicine, Louisville, KY 40202, USA; eakorte@gmail.com (E.A.K.); michael.merchant@louisville.edu (M.L.M.); jon.klein@louisville.edu (J.B.K.); michelle.barati@louisville.edu (M.T.B.); ami.joglekar@gmail.com (A.J.); daniel.wilkey@louisville.edu (D.W.W.); kenneth.mcleish@louisville.edu (K.R.M.); david.powell@louisville.edu (D.W.P.); 2Robley Rex Veterans Affairs Medical Center, Louisville, KY 40206, USA; 3Pathology Department, Norton Children’s Hospital, Louisville, KY 40202, USA; susan.coventry@nortonhealthcare.org (S.C.); Jessica.Hata@nortonhealthcare.org (J.H.); 4Department of Medicine, The Ohio State University, Columbus, OH 43210, USA; Brad.Rovin@osumc.edu; 5Center for Autoimmune Genomics and Etiology, Department of Pediatrics, Cincinnati Children’s Hospital Medical Center, College of Medicine, University of Cincinnati, Cincinnati, OH 45267, USA; john.harley@cchmc.org (J.B.H.); Bahram.Namjou@cchmc.org (B.N.-K.); 6US Department of Veterans Affairs Medical Center, Cincinnati, OH 45220, USA

**Keywords:** autoantibodies, glomerulonephritis, lupus nephritis, systemic lupus erythematosus, target antigens

## Abstract

Kidney involvement in systemic lupus erythematosus (SLE)—termed lupus nephritis (LN)—is a severe manifestation of SLE that can lead to end-stage kidney disease (ESKD). LN is characterized by immune complex deposition and inflammation in the glomerulus. We tested the hypothesis that autoantibodies targeting podocyte and glomerular cell proteins contribute to the development of immune complex formation in LN. We used Western blotting with SLE sera from patients with and without LN to identify target antigens in human glomerular and cultured human-derived podocyte membrane proteins. Using liquid chromatography-tandem mass spectrometry (LC-MS/MS), we identified the proteins in the gel regions corresponding to reactive bands observed with sera from LN patients. We identified 102 proteins that were present in both the podocyte and glomerular samples. We identified 10 high-probability candidates, including moesin, using bioinformatic analysis. Confirmation of moesin as a target antigen was conducted using immunohistochemical analysis (IHC) of kidney biopsy tissue and enzyme-linked immunosorbent assay (ELISA) to detect circulating antibodies. By IHC, biopsies from patients with proliferative lupus nephritis (PLN, class III/IV) demonstrated significantly increased glomerular expression of moesin (*p* < 0.01). By ELISA, patients with proliferative LN demonstrated significantly increased antibodies against moesin (*p* < 0.01). This suggests that moesin is a target glomerular antigen in lupus nephritis.

## 1. Introduction

Systemic lupus erythematosus (SLE) is a complex autoimmune disease characterized by the loss of self-tolerance and development of autoantibodies that can be initiated by a combination of genetic, environmental, hormonal, and immunoregulatory factors [1]. One of the hallmarks of SLE is the abnormal clearance of apoptotic cells resulting in the generation of anti-nuclear antibodies (ANAs), which include antibodies to double-stranded DNA (dsDNA) [1]. Antibodies to dsDNA are very specific for SLE, and serum levels often associate with disease activity [2]. Approximately 50% of patients with SLE develop glomerular injury, termed lupus nephritis (LN), due to the deposition of immune complexes and subsequent inflammatory response [3]. LN is further categorized into classes based on the location of immune deposits, corresponding inflammation, and percent of glomeruli involved in the World Health Organization (WHO) LN classification system and the newer International Society of Nephrology/Renal Pathology Society (ISN/RPS) classification system [4]. Most patients with clinically relevant LN have a proliferative form (class III or IV; PLN), while a minority of patients (20–40%) develop pure membranous LN (class V, MLN) [5].

The target antigens of autoantibodies in LN, which lead to glomerular immune com-plex deposition and produce disease, are not well defined. Multiple theories have been proposed to explain immune complex deposition in the glomerulus. While they are separate processes, they are not mutually exclusive, and all may contribute to the development of LN. First, immune complexes which have formed in the circulation deposit directly into the glomerular vasculature. Second, antibodies bind to an antigen (such as nucleosomes or C1q) that is “planted” into the glomerulus. Finally, autoantibodies may bind to an endogenous glomerular antigen, resulting in “in situ” immune complex formation [6]. There is a correlation between anti-dsDNA antibody levels and the development of renal disease, and anti-dsDNA antibodies have been identified in glomerular immune complexes [6,7,8,9,10]. However, there are additional antibodies involved in the development of LN. A study evaluating eluted IgG from LN glomeruli demonstrated reactivity to dsDNA in only 25% of patients; quantitatively, those antibodies reacting to dsDNA represented a small fraction of total IgG, accounting for less than 1% of the eluted IgG [7]. This suggests that autoantibodies targeting non-nuclear antigens play a significant role in LN. The identification of target glomerular antigens in LN will lead to an improved understanding of LN pathogenesis and provide better diagnostic biomarkers. The current study was designed to detect autoantibodies to glomerular antigens that were differentially expressed in proliferative vs. membranous LN.

## 2. Materials and Methods

### 2.1. Study Subjects

LN sera were obtained from the Ohio SLE Study cohort [11] and the Lupus Family Registry and Repository [12]. Subjects from the Ohio SLE Study cohort were in renal flare at time of sample collection, as defined by proteinuria and/or elevation in serum creatinine [11]. SLE samples without nephritis (lupus controls, LC) were obtained from the Lupus Family Registry and Repository [12]. Normal controls (NC) were obtained from healthy adult volunteers at the University of Louisville (Louisville, KY, USA). There were 45 research subjects in total, including 10 lupus controls (LC), 10 PLN (LN class III or class IV), 15 MLN (LN class V), and 10 normal controls (NC). SLE subjects met ACR diagnostic criteria for SLE. LC subjects never had a diagnosis of LN and did not have clinical evidence of kidney involvement with SLE. PLN subjects had class III or IV LN lesions, as determined by kidney biopsy. MLN subjects had class V lesions on kidney biopsy in the absence of class III or IV lesions. Clinical characteristics from the time of sample collection were available for all 10 PLN subjects and 5 MLN subjects, as shown in Table 1.

Deidentified, remnant kidney biopsy tissue from PLN and MLN patients were obtained from collaborating pathologists. The Internal Review Board at all institutions approved sample donation and sharing. Normal human kidney tissue was obtained from deceased donor kidneys not suitable for transplantation (courtesy of Kentucky Organ Donor Affiliates, Louisville, KY, USA).

### 2.2. Glomerular and Podocyte Protein Extracts

Human glomerular and cultured human-derived podocyte protein lysates were prepared as previously described [13]. Briefly, size-separating stainless steel sieves were utilized to isolate glomeruli from the cortex [14]. Light microscopy verified glomerular purity at >90%. Isolated glomeruli were sonicated to break up matrix and placed in lysis buffer containing protease and phosphatase inhibitor [13]. Contaminating IgG was removed to prevent any cross reactivity with anti-human IgG, and the depletion was validated by Western blot [13].

Podocyte protein lysates were generated from a line of conditionally immortalized human podocytes (a kind gift from Moin Saleem) [15]. Podocytes were cultured and allowed to differentiate on collagen-coated plates at 37 °C, as previously described [13]. Sequential centrifugation of the homogenate was utilized to separate membrane and cytosol as previously described [13].

### 2.3. Western Blot Analysis 

Human glomerular and podocyte membrane protein samples were separated by using 5–15% gradient gels with sodium dodecyl sulphate–polyacrylamide gel electrophoresis (SDS/PAGE). Proteins were transferred to polyvinylidene difluoride (PVDF) or nitrocellulose membranes. Gel electrophoresis was conducted under non-reducing conditions for the podocyte protein extracts. Initial gel electrophoresis of the glomerular extracts was conducted under both non-reducing and reducing conditions, with 2-mercaptoethanol (BME) as the reducing agent. The Western blots demonstrated more reactive bands under reducing conditions, and thus were used for the final analysis. The membranes were incubated with individual subject sera (1:100–1:200) for glomerular extracts and pooled sera (1:10,000) for podocyte membrane extracts overnight at 4 °C. Following sera incubation and washing, membranes were incubated with HRP-conjugated anti-human IgG followed by chemoluminescent substrate, as previously described [13]. Final images were developed on Bio-Rad ChemiDoc (Bio-Rad, Hercules, CA, USA) or with film.

### 2.4. Mass Spectrometry Analysis

Candidate target antigens were identified from human glomerular extracts using an SDS-PAGE (”gel-C”) mass spectrometry approach that used previously described methods [16,17,18]. Data were collected in a data-dependent fashion using an Nth Order Double Play with ETD Decision Tree method created in Xcalibur v2.2. Database Searching (Thermo Scientific, Waltham, MA, USA) was completed as previously described [19].

### 2.5. Enzyme-Linked Immunosorbent Assay

Recombinant moesin (MSN) was expressed and purified using a His-tagged bacterial expression vector obtained from Arizona State University’s plasmid repository (ASU Biodesign Institute, Tempe, AZ, USA) [20]. A sandwich ELISA technique was utilized using a previously described technique [13]. Briefly, polyclonal moesin antibody (sc-6410, Santa Cruz Biotechnology, Inc., Dallas, TX, USA) was incubated at 4 °C overnight on a standard 96-well plate to serve as a capture antibody. Following routine washing and blocking, recombinant moesin (0.2 µg per well) was added and incubated for 2 h. Diluted sera (1:200) and sera-free controls were added in duplicate and incubated for 1 h, followed by washing. HRP-conjugated anti-human IgG (sc-2769, Santa Cruz Biotechnology, Inc., Dallas, TX, USA) was added to each well at a concentration of 1:1000 for 30 min, followed by washing. Finally, 3,3′,5,5′-tetramethylbenzidine substrate (Thermo Scientific, Waltham, MA, USA) was added for 15 min and allowed to develop while protected from light. Sulfuric acid (Thermo Scientific, Waltham, MA, USA) was used to stop the reaction, and the optical density (OD) was immediately measured at 450 nM and normalized at 570 nM. The OD of the sera-free control was subtracted from the measured OD of all samples to reduce non-specific reactivity.

### 2.6. Immunohistochemistry

Deidentified, formalin-fixed paraffin-embedded (FFPE) renal tissue from patients with PLN and MLN was provided by collaborating pathologists. Immunohistochemical analysis (IHC) of paraffin-embedded renal biopsy sections was performed as previously described [21]. Briefly, sections were incubated overnight at 4 °C with goat polyclonal moesin antibody (sc-6410, Santa Cruz Biotechnology, Inc., Dallas, TX, USA) or rabbit monoclonal moesin antibody at 1:50 (Abcam 52490, Abcam, Cambridge, UK), followed by incubation with biotinylated donkey anti-goat secondary antibody at 1:100 (sc-2042, Santa Cruz Biotechnology, Inc., Dallas, TX, USA) or biotinylated anti-rabbit secondary antibody at 1:500 (BA-1000,Vector Labs, Burlingame, CA, USA). Each section was then incubated with avidin-biotin-complex substrate (PK-4000, Vector Labs, Burlingame, CA, USA). Following incubation, each sample was exposed to DAB (SK-4100, Vector Labs, Burlingame, CA, USA) for one minute, followed by immediate washing, mounting, and imaging. Digital images were acquired with a Q Color 5 camera attached to an Olympus BX51 microscope (Olympus Corporation, Center Valley, PA, USA) using Image-Pro 6.2 software (Media Cybernetics, Silver Spring, MD, USA). Moesin glomerular tuft immunostaining was quantified with Image-Pro 6.2 software (Media Cybernetics, Silver Spring, MD, USA) on images captured with a 40× objective. The percentage of immunostaining in each glomerular tuft was determined by the ratio of immunostained area to total glomerular tuft surface area.

### 2.7. Statistical Analyses

Statistical analysis for the ELISA and IHC quantification was performed using one-way analysis of variance (ANOVA) with Tukey HSD post-test to verify differences between groups. Harmonic mean was used in the IHC analysis due to an unequal group size (SPSS V 27, IBM, Armonk, NY, USA).

## 3. Results

### 3.1. Identification of Candidate LN Target Antigens

Sera from patients with PLN (class III/IV LN), MLN (class V LN), and SLE patients without renal disease (LC) were immunoblotted against isolated human glomerular or podocyte protein extracts. LN sera produced multiple reactive bands, but bands at approximately 50–60 kDa in the plasma membrane fraction prepared from culture human-derived podocytes (Figure 1a) and approximately 40–55 kDa in glomerular extracts (Figure 1b) appeared to be more specific to LN, which we previously reported [13]. Our prior analysis of the gel region between 50–60 kDa and 40–60 kDa identified 294 proteins from podocyte membrane lysate and 310 proteins from human glomerular lysate, respectively; 102 proteins were common in the two groups (Figure 1c) [13]. The differences in molecular weight range for reactive bands in these two samples could be due to differences in the overall composition of the lysate, differences in protein modifications, or fragmentation. These postulates are supported by a high number of identification overlap. The overlap also suggests that many of the reactive proteins in the glomerular homogenate are podocyte and/or membrane proteins. We chose to focus on known membrane-associated proteins because target antigens in other immune-mediated glomerular diseases, such as primary membranous nephropathy, are membrane associated [16,22]. We used the Gene Ontology database to identify 36 membrane-associated proteins [13,23]. A PubMed search of these proteins identified 10 proteins previously reported to participate in glomerular diseases (Table 2).

### 3.2. Autoantibodies to Moesin Are Elevated in PLN, Not MLN

Moesin was chosen for further study based on its reported contribution to glomerulonephritis, association with the slit diaphragm, participation in actin remodeling, and presence in glomerular mesangial cells, endothelial cells, and epithelial cells [32,33,39,40,41]. Additionally, antibodies eluted from LN glomeruli and present in LN sera reacted to moesin in a previous study [35]. To validate the presence of anti-moesin autoantibodies, we developed a sandwich ELISA protocol to reduce non-specific binding to contaminants typically present in recombinant protein preparations. Sera at 1:200 from 10 PLN, 10 MLN, 10 LC, and 10 NC subjects were analyzed in duplicate. Clinical data at the time of sample collection are available for all PLN samples, and were collected during known renal flare, defined by elevated protein and/or creatinine from baseline [11]. Table 1 shows known clinical data at the time of sample collection, when available.

Figure 2 shows that sera from PLN subjects demonstrated a significantly higher IgG reactivity to moesin than those from MLN, LC, and NC subjects. One-way analysis of variance (ANOVA) comparing individual reactivity to anti-moesin demonstrated a *p*-value of <0.01 and Tukey HSD verified statistical differences among groups comparing PLN to MLN, LC, and NC subjects (*p* < 0.01). Thus, sera from patients with PLN contained autoantibodies that recognize moesin, while sera from patients with pure MLN or SLE without LN did not differ from normal subjects.

### 3.3. Biopsies from Patients with PLN Demonstrate Increased Glomerular Expression of Moesin

Known target antigens in glomerular diseases are frequently characterized by increased glomerular expression on kidney biopsy [22,42,43]. To further assess moesin as a target antigen in PLN, immunohistochemical (IHC) staining for moesin was performed on human kidney biopsies from patients with PLN (*N* = 8), MLN (*N* = 3), and normal controls (NC, *N* = 6). Moesin IHC demonstrated strong global granular capillary loop staining in PLN. There was also staining in podocytes and parietal epithelial cells (Figure 3a,b). Moesin staining in MLN demonstrated faint capillary loop staining in a pattern similar to NC (Figure 3c,d). NC also showed staining of podocytes and parietal epithelial cells (Figure 3d). Glomerular moesin staining was quantified and normalized to glomerular area. Glomerular staining was significantly increased in the PLN samples by one-way ANOVA (*p* = 0.001) and there were significant differences between PLN and MLN (*p* = 0.006) as well as PLN and NC (*p* = 0.003), as verified with Tukey HSD test (Figure 3e).

## 4. Discussion

The present study utilized a proteomic approach to identify moesin as a candidate target glomerular antigen in LN. We validated circulating anti-moesin antibodies using the serum ELISA of individual patients and demonstrated that patients with PLN, but not MLN, had significantly increased circulating IgG to moesin. We also found significantly increased glomerular expression of moesin by IHC. Our study suggests moesin may be added to a list of glomerular antigens, including alpha actinin, alpha enolase, annexin A1, annexin A2, heparan sulfate, and laminin, previously identified as potential targets of autoantibodies in LN [13,25,28,44,45,46,47,48,49,50,51]. Our findings are further supported by the work of Bonanni et al., who used a similar approach, immunoblotting sera and eluted glomerular immunoglobulin from LN patients against human podocyte extracts separated by 2D gel electrophoresis [35]. Using MALDI and LC/MS of the reactive spots, they identified multiple podocyte antigens—including ezrin/moesin—as targets for both eluted glomerular antibodies and circulating autoantibodies [35].

Moesin is part of the ezrin–radixin–moesin (ERM) complex involved in canonical Rho GTPase signaling, which is essential for actin cytoskeleton remodeling, cell adhesion, and motility. Anti-moesin antibodies have been identified in several autoimmune diseases, including anti-neutrophil cytoplasmic antibody (ANCA) vasculitis, acquired aplastic anemia, polyarteritis nodosa, anti-phospholipid antibody syndrome, and Sjogren’s syndrome [33,52,53,54,55]. The identification of anti-moesin antibodies in a spectrum of autoimmune diseases may suggest cross-reactive antibody binding rather than unique autoantibodies to moesin. However, that does not diminish the importance of this identification, as it may provide additional insights into the pathogenesis of the disease. In ANCA vasculitis, anti-MPO antibodies cross-reactive to moesin in glomerular endothelial cells lead to endothelial cell activation and increased glomerular moesin expression [56].

Current SLE autoantibodies, including antibodies to dsDNA, are not uniformly associated with nephritis and do not distinguish between histologic classes in LN. Although the current study does not distinguish whether anti-moesin antibodies were the result of tissue-specific autoantibodies or cross-reacting anti-dsDNA antibodies, the data suggest that antibodies that react with moesin are associated with LN class III and IV (PLN). 

Based on the current studies, we cannot conclude whether the antibodies to moesin are pathogenic or an epiphenomenon that occurs in the setting of tissue inflammation and damage. Autoantibodies to moesin may not be the initiating event, but may contribute to ongoing inflammation. Deng et al. demonstrated that high mobility group box-1 (HMGB1), a damage-associated molecular pattern molecule (DAMP), can lead to increased expression of moesin on glomerular endothelial cells. They propose that the increased expression leads to autoantibody binding, which leads to further endothelial cell activation, increased permeability, and disruption of the vascular barrier in crescentic glomerulonephritis [57]. We demonstrated significantly increased glomerular moesin expression in biopsies from PLN, but not MLN or NC. In other glomerular diseases, such as primary and secondary membranous nephropathy, increased expression of glomerular antigens has been seen [22,42,43]. In primary membranous nephropathy, antibodies to glomerular antigens lead to in situ immune complex deposition [22,42].

We did not show direct co-localization of moesin with immune deposits, and thus cannot definitively say that antibodies targeting moesin are key players in immune complex formation in PLN. However, tissue-targeted antibodies may have direct effects on glomerular cells without immune complex formation. For example, Manson et al. demonstrated that IgG from LN sera, but not from the sera of SLE patients without nephritis, had direct effects on tyrosine phosphorylation in podocyte proteins [58]. As noted above, anti-moesin antibodies have direct effects on glomerular endothelial cells in the absence of immune complex formation in ANCA vasculitis [56,57]. Further studies are needed to identify whether anti-moesin antibodies correlate with disease activity, and whether anti-moesin antibodies are pathogenic. Our studies suggest that quantitative serum anti-moesin antibodies and IHC for glomerular moesin staining may be useful biomarkers to identify patients with LN class III and IV (PLN), but not pure class V LN (MLN).

## Figures and Tables

**Figure 1 jcm-10-00793-f001:**
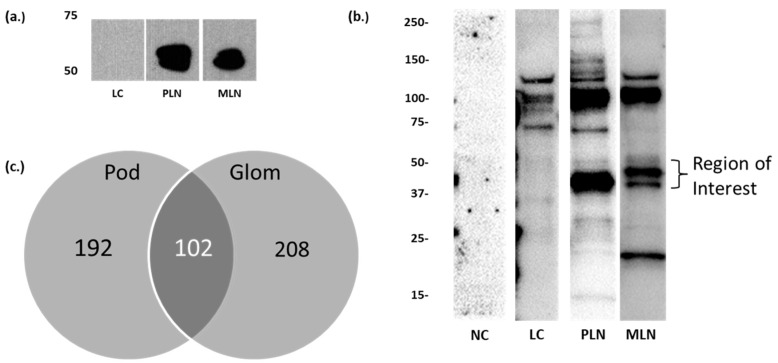
Identification of candidate antigens. (**a**) Western blot of pooled normal control (NC), lupus control (LC), proliferative lupus nephritis (PLN), and membranous lupus nephritis (MLN) sera (1:10,000 dilution) against human podocyte membrane proteins showing reactive band at approximately 50–60 kDa against 20 µg of podocyte membrane proteins in subjects with LN (previously reported in [13]). (**b**) Representative Western blots of NC, LC, PLN, and MLN sera (1:100) against 20 µg human glomerular extract (HGE) showing a region of interest between 40 and 55 kDa. (**c**) Identification of 102 overlapping glomerular (Glom) and podocyte (Pod) proteins using Scaffold software ver4.3.4 (Proteome Software, Portland, OR, USA).

**Figure 2 jcm-10-00793-f002:**
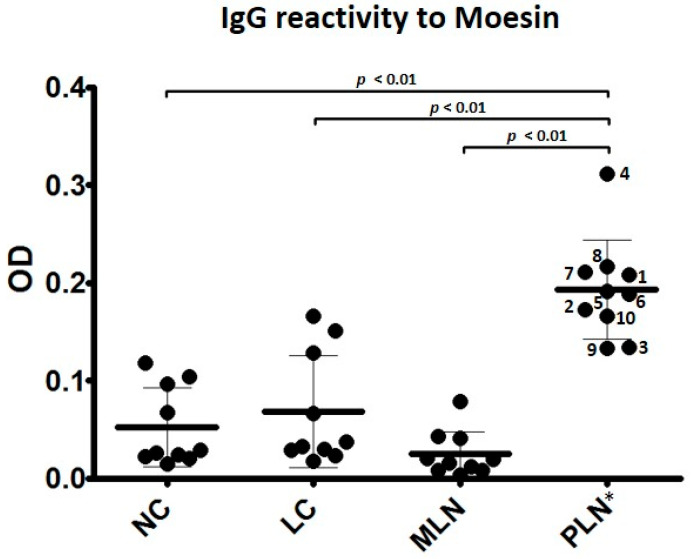
Anti-moesin IgG ELISA. Sera at 1:200 dilution from 10 PLN, 10 MLN, 10 LC, and 10 NC subjects were analyzed in duplicate. NC subjects demonstrated a mean optical density (OD) value of 0.053, LC subjects demonstrated a mean OD value of 0.067, MLN subjects demonstrated a mean OD value of 0.025, and PLN subjects demonstrated a mean OD value of 0.192. One-way ANOVA demonstrated significance with a *p*-value < 0.01. Tukey HSD test demonstrated a statistical difference among groups comparing PLN to MLN, LC, and NC subjects (*p* < 0.01). Error bars demonstrate standard error of the mean (SEM).

**Figure 3 jcm-10-00793-f003:**
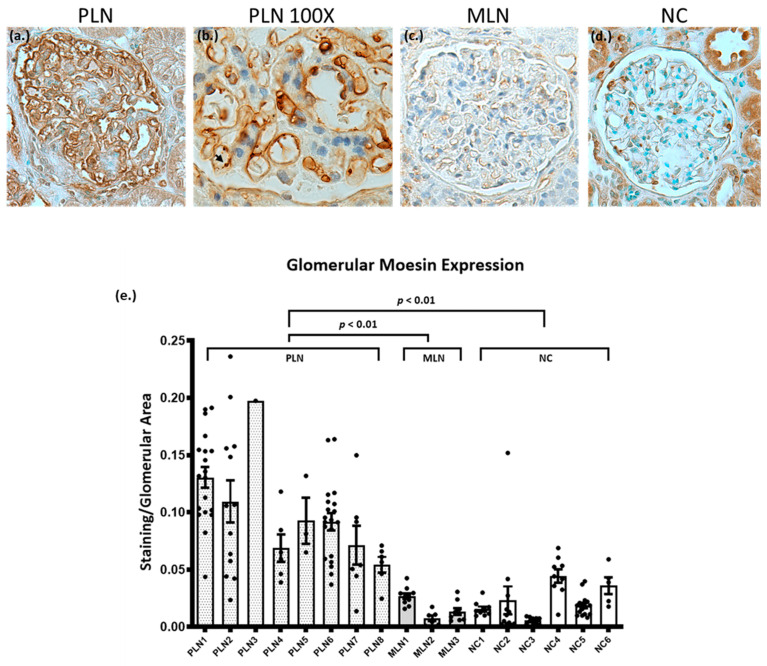
Immunohistochemistry demonstrating increased glomerular moesin staining in PLN. (**a**) Moesin IHC of PLN glomerulus (representative sample) demonstrates strong global granular capillary loop staining. Moesin also stains the cytoplasm of Bowman’s capsule parietal cells and podocytes. (**b**) Moesin IHC of PLN glomerulus at 100×, demonstrating strong capillary loop staining with arrow showing strong staining on luminal surface and endothelial cell staining (**c**) Moesin IHC of MLN glomerulus (representative sample) demonstrates very light capillary loop staining, similar to what was seen in normal control. (**d**) Moesin IHC of an NC glomerulus (representative sample) demonstrates very light capillary loop staining, rare, nonspecific nuclear and cytoplasmic staining of podocytes. Moesin also stains the cytoplasm of Bowman’s capsule parietal cells. (**e**) Quantitative analysis of glomerular moesin staining normalized to glomerular area. Each point represents an individual glomerulus. LN samples had mean staining/glomerular area of 0.1304, 0.1094, 0.1974, 0.06854, 0.0926, 0.0884, 0.0713, and 0.0541; MLN samples had mean staining/glomerular area of 0.0267, 0.0075, and 0.0129. NC samples had mean staining/glomerular area of 0.0156, 0.0231, 0.0057, 0.0443, 0.0355, and 0.0360. There was a statistically significant difference between LN, MLN, and NC glomeruli using ANOVA (*p* = 0.001), and the Tukey post-test demonstrates a significant difference between PLN and MLN (*p* = 0.006) and between PLN and NC (*p* = 0.003), but comparison of MLN and NC indicated no significant difference (*p* = 0.89). Error bars demonstrate SEM.

**Table 1 jcm-10-00793-t001:** Clinical characteristics of lupus nephritis (LN) subjects.

Subject	Sex	Age	Race	WHO Class *	Creatinine mg/dL	eGFR **	UPCR g/g	Anti-dsDNA ab	C3	C4
1	F	56	W	4	2.5	21	2.1	NA	102	NA
2	F	46	W	4	0.9	77	2.91	+	64	7
3	F	44	AA	4	1.6	45	4.8	NA	71	11
4	F	36	AA	3,4	1.1	75	2.3	NA	72	22
5	F	20	AA	4	0.5	161	NA	NA	61	3
6	F	23	AA	4	1	92	NA	NA	98	13
7	F	29	Asian	4	0.8	100	2.2	NA	74	7
8	F	19	AA	4	4.34	16	NA	NA	43	20
9	F	21	W,H	4	1.82	39	NA	+	68	8
10	F	34	W	4	1.53	44	1.36	NA	81	13
11	F	26	AA	5	1.6	51	16.5	+	67	17
12	F	24	AA	5	0.9	104	4.4	+	65	15
13	F	60	W	5	0.68	95	0.97	NA	138	20
14	F	37	W	5	0.63	115	4.2	+	63	11
15	F	22	W	5	1.43	52	1.1	−	93	21

* WHO Class V only including “pure membranous” without evidence of focal or diffuse proliferative lesions. ** eGFR = estimated glomerular filtration rate using CKD-EPI Equation. F = female; W = white; AA = African American; H = Hispanic; UPCR = Urine Protein to Creatinine Ratio; NA = Not Available ab = antibody; C3 = Complement component 3; C4 = Complement component 4.

**Table 2 jcm-10-00793-t002:** Candidate antigens.

Protein	Gene Name	GN	Slit Diaphragm Protein ^1^
Actin-related protein 3	ACTR3	[24]	
Alpha-enolase	ENO1	[25,26]	
Alpha-parvin	PARVA	[27]	+
Annexin A2	ANXA2	[13,28]	+
Ezrin	EZR	[29,30]	+
Isoform 2 of basigin	BSG	[31]	
Moesin	MSN	[32,33,34,35]	+
Myosin-9	MYH9	[36]	+
V-type proton ATPase subunit B, brain isoform	ATP6V1B2	[37]	
60 kDa heat shock protein, mitochondrial	HSPD1	[38]	

GN = Glomerulonephritis; ^1^ Slit diaphragm proteins [39]; + = positively identified.

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
