# Peer review of "Patients with Proliferative Lupus Nephritis Have Autoantibodies That React to Moesin and Demonstrate Increased Glomerular Moesin Expression"

_jcm, 2021, doi:10.3390/jcm10040793_

Round 1

Reviewer 1 Report

Comments to the Authors,

     In the present study, authors utilized a proteomic approach to identify a target antigen for patients with lupus nephritis (LN), and they demonstrated that moesin on glomerular basement membrane (GBM) could be a candidate for a novel glomerular antigen in patients with proliferative LN, not membranous LN (MLN). The results obtained from this study could be relevant and intriguing for physicians who treated LN patients. Authors clearly summarized the results, and discussion was also simply described in order to grasp the essential information. However, some points remain still unconvincing and elusive. In addition, the method and discussion in the present study were similar to those in the recent reports (Sethi S et al. JASN 30: 1123–1136) (Sethi S et al. Kidney Int 2020: 97;163–174). Furthermore, as authors mentioned, Bonanni et al. already reported that ezrin/moesin on podocyte has a potential as a target antigen for antibody eluted from glomeruli and circulating autoantibodies in LN.

So, please consider to address my suggested questions.

Major revision:

1, Authors defined glomerular moesin is a novel target antigen in patients with PLN in the section of title and conclusion of abstract. However, I have to be suspicious. Indeed, authors demonstrated the up-regulation of moesin on GBM in only three patients with PLN by the method with immunohistochemistry (IHC). Meanwhile, they did not show the presence of immunoglobulin targeting moesin in those three patients. To conclude the moesin as a target antigen for immunoglobuline on glomerulus, authors need to show double IHC staining of IgG and moesin by using same paraffin section. If impossible, at least, please show the figures of IHC staining for moesin and IgG, separately with two consecutive paraffin sections. Authors need to evaluate whether the distribution of moesin and immunoglobulin is matching on the same glomerulus in patients with PLN. Otherwise it might be difficult to define moesin as a glomerular target antigen in patients with PLN.       

2, Authors mentioned that moesin was immanent in not only GBM or capillary loop but also mesangial or endothelial area (Page 5, Line 177). If so, please consider to discuss the reason why the up-regulation of glomerular moesin was detected on GBM, not endothelial and mesangial area in patients with PLN who have typical immune deposition on mesangial or endothelial area. Basically, I am wondering why the expression of glomerular moesin was increased in GBM side in PLN (as a result of shedding such as PLA2R on GBM in primary MN?). I will accept such finding without any discomfort if the glomerular up-regulation of moesin was detected in patients with MLN in which subepithelial immune deposit is detected.

3, Actually, authors directly extracted glomerular protein from patients with LN, meanwhile they did not directly isolate podocyte from glomerulus of patients. Authors seem to utilize the immortalized human podocyte cell line. Thus, I am afraid that the influence of its immortality may modify the results in the present study. In the Figure 1, bands on the membrane for western blotting were different according to the samples. In the samples of podocyte (cell line), strong band between 50 and 75 kDa was detected, whereas such band between 50 and 75 kDa was not detected in the samples of glomerular extracts which will contain the component of podocyte. In the manuscript, authors simply explained the reason while referring their previous report, which may be confused and elusive for readers. Please add an additional explanation the reason why the bands which reacted sera from LN patients were different between human cell line and directly extracted glomerular protein.

4, As authors mentioned, serum anti-moesin antibodies (Ab) have been identified in several autoimmune diseases including ANCA-related GN, which implied that the anti-moesin Ab might not be specific in patients with LN. I presumed that the anti-moesin Ab, recognized in the present study, may be actually autoantibody itself such as anti-dsDNA Ab and anti-SmAb, and ANCA. In other word, reaction detected in ELISA may reflect the cross-reaction of autoantibody for another antigen in patients with PLN. In general, around 40-50% of MLN patients did not show low serum complement and high anti-DNA Ab titer at presentation. Thereby, serum from MLN patients in the present study might not response to the moesin-antigen layered on the ELISA plate because of the defect of serum autoantibody in MLN. Pleas comment and discuss above mentioned hypothesis and possibility.  

5, Please show the results regarding the baseline characteristics between the patients with PLN and MLN. The results should be simply presented and should consist of baseline data (age, gender, and so on), renal function (sCr, eGFR), urinary findings (proteinuria and degree of hematuria), immunoserological results (complement, ANA, anti-dsDNA, and so on), and treatment modalities (dose of steroid therapy and other immunosuppressive therapy).

6, Please increase the sample number for IHC analysis in Figure 3. Each 3 samples are too small to discuss or conclude. In addition, authors need to show the figures of IHC staining for moesin in patients with ML as well as PLN.

7, In the discussion section, authors mentioned as follow; Further studies are needed to identify whether anti-moesin antibodies correlate with disease activity. I strongly agree with your suggestion and I am so interested in the relationship between the serum anti-moesin Ab or glomerular moesin expression and disease activity of LN. Please show the data regarding the correlation between such Ab or glomerular expression and complement or anti-dsDNA Ab levels in patients with LN, although the number of enrolled patients was small.

Minor revision:

1, In the abstract, authors frequently used the term “podocyte and glomerular extracts”. However, authors seemed not to extract or isolate podocyte. They seem to adopt cell line of podocyte. Please consider to change its description.

2, In the introduction, authors frequently mentioned as following “anti-dsDNA serum levels”. To be precise, it should be corrected as anti-dsDNA antibody. Please make sure all parts of manuscript when the term ds-DNA was described.

3, In the categorization of PLN or MLN, authors described LN class III, IV, and V. Is this classification based on the ISN/RPS 2003, correct? Please mention the famous classification for LN in the manuscript. In addition, do authors include the patients with combined class III/V + V as PLN? If not, please clarify that LN patients in combined class III/IV +V were not contained in the subjects of PLN.  

4, Authors need to simply explain the method of statistical analysis in the section of material method.

5, Please show the unit of the graph in Figure 3c. Authors mentioned as follow; Percent of immunostaining in each glomerular tuft was determined by the ratio of immunostained area to total glomerular tuft surface area. I think that staining area in representative photo in Figure 3a is over 25%?

Author Response

Major revision

1, Authors defined glomerular moesin is a novel target antigen in patients with PLN in the section of title and conclusion of abstract. However, I have to be suspicious. Indeed, authors demonstrated the up-regulation of moesin on GBM in only three patients with PLN by the method with immunohistochemistry (IHC). Meanwhile, they did not show the presence of immunoglobulin targeting moesin in those three patients. To conclude the moesin as a target antigen for immunoglobuline on glomerulus, authors need to show double IHC staining of IgG and moesin by using same paraffin section. If impossible, at least, please show the figures of IHC staining for moesin and IgG, separately with two consecutive paraffin sections. Authors need to evaluate whether the distribution of moesin and immunoglobulin is matching on the same glomerulus in patients with PLN. Otherwise it might be difficult to define moesin as a glomerular target antigen in patients with PLN.

We appreciate the feedback from the reviewer and have adjusted the title of the manuscript and clarified our conclusions with additional discussion.

2, Authors mentioned that moesin was immanent in not only GBM or capillary loop but also mesangial or endothelial area (Page 5, Line 177). If so, please consider to discuss the reason why the up-regulation of glomerular moesin was detected on GBM, not endothelial and mesangial area in patients with PLN who have typical immune deposition on mesangial or endothelial area. Basically, I am wondering why the expression of glomerular moesin was increased in GBM side in PLN (as a result of shedding such as PLA2R on GBM in primary MN?). I will accept such finding without any discomfort if the glomerular up-regulation of moesin was detected in patients with MLN in which subepithelial immune deposit is detected.

We have included additional staining of MLN (pure class V) which does not show increased glomerular expression of moesin and suggests this is unique to class III/IV LN. Class III/IV lesions include both subendothelial and mesangial deposits.  In class III/IV lesions we saw a pattern of significantly increased capillary staining which may suggest there is increased expression of moesin by endothelial cells, but not mesangial cells in PLN.  We have added a 100X image to show detail of endothelial staining. Of note, there was also staining in podocytes and parietal epithelial cells. The increased staining may be a result of direct injury to these cells (please see added discussion).

3, Actually, authors directly extracted glomerular protein from patients with LN, meanwhile they did not directly isolate podocyte from glomerulus of patients. Authors seem to utilize the immortalized human podocyte cell line. Thus, I am afraid that the influence of its immortality may modify the results in the present study. In the Figure 1, bands on the membrane for western blotting were different according to the samples. In the samples of podocyte (cell line), strong band between 50 and 75 kDa was detected, whereas such band between 50 and 75 kDa was not detected in the samples of glomerular extracts which will contain the component of podocyte. In the manuscript, authors simply explained the reason while referring their previous report, which may be confused and elusive for readers. Please add an additional explanation the reason why the bands which reacted sera from LN patients were different between human cell line and directly extracted glomerular protein. 

To clarify, the glomerular protein extract was from normal human kidneys that were turned down for transplant. We were trying to identify antibody reactivity to glomerular proteins and used sera from LN patients to identify reactivity to extracted proteins from normal tissue. This is outlined in the methods section.

We agree that the difference in the molecular weight ranges for the reactive bands of interest for the two samples should be explained better. More precisely the molecular weight range for the reactive bands of interest with the cultured podocyte sample is 50-60 kDa and the range for the two reactive bands of interest with the glomerular homogenate is 40-55.  These were ranges that were cut from respective gels for mass spectrometry analysis. This is clarified in the results section 3.1.  We have also added an explanation for the difference in molecular weight ranges for the two different samples. 

4, As authors mentioned, serum anti-moesin antibodies (Ab) have been identified in several autoimmune diseases including ANCA-related GN, which implied that the anti-moesin Ab might not be specific in patients with LN. I presumed that the anti-moesin Ab, recognized in the present study, may be actually autoantibody itself such as anti-dsDNA Ab and anti-SmAb, and ANCA. In other word, reaction detected in ELISA may reflect the cross-reaction of autoantibody for another antigen in patients with PLN. In general, around 40-50% of MLN patients did not show low serum complement and high anti-DNA Ab titer at presentation. Thereby, serum from MLN patients in the present study might not response to the moesin-antigen layered on the ELISA plate because of the defect of serum autoantibody in MLN. Pleas comment and discuss above mentioned hypothesis and possibility.  

We agree this is an important point. The reactivity of IgG from patients with proliferative lupus nephritis may represent cross reactive binding of anti-dsDNA ab.  However, many of our MLN and LC subjects had h/o anti-dsDNA ab positivity. This may suggest that some forms of anti-dsDNA are more nephritogenic and identifying glomerular antigens that circulating autoantibodies can bind to (whether unique or cross reactive) offers insights into the pathogenesis of LN and also may serve as a better diagnostic biomarker than anti-dsDNA ab alone.

5, Please show the results regarding the baseline characteristics between the patients with PLN and MLN. The results should be simply presented and should consist of baseline data (age, gender, and so on), renal function (sCr, eGFR), urinary findings (proteinuria and degree of hematuria), immunoserological results (complement, ANA, anti-dsDNA, and so on), and treatment modalities (dose of steroid therapy and other immunosuppressive therapy).  

We have added clinical data from time of sample collection, when available (table 1).  

6, Please increase the sample number for IHC analysis in Figure 3. Each 3 samples are too small to discuss or conclude. In addition, authors need to show the figures of IHC staining for moesin in patients with ML as well as PLN.  

We have increased the number of samples and added MLN samples. The results remain statistically significant.

7, In the discussion section, authors mentioned as follow; Further studies are needed to identify whether anti-moesin antibodies correlate with disease activity. I strongly agree with your suggestion and I am so interested in the relationship between the serum anti-moesin Ab or glomerular moesin expression and disease activity of LN. Please show the data regarding the correlation between such Ab or glomerular expression and complement or anti-dsDNA Ab levels in patients with LN, although the number of enrolled patients was small.

We have added a clinical data table.  While we do not have detailed clinical data on all subjects, we do have data on all 10 PLN subjects that demonstrated increased circulating IgG reactive to moesin.  All of these subjects were in a renal flare as defined by increase in proteinuria and/or creatinine from baseline.

Minor revision:

1, In the abstract, authors frequently used the term “podocyte and glomerular extracts”. However, authors seemed not to extract or isolate podocyte. They seem to adopt cell line of podocyte. Please consider to change its description.

We now note that human-derived cultured podocytes were used for the podocyte samples in the abstract, page 1, and line 6.  

2, In the introduction, authors frequently mentioned as following “anti-dsDNA serum levels”. To be precise, it should be corrected as anti-dsDNA antibody. Please make sure all parts of manuscript when the term ds-DNA was described. 

These changes are made throughout the manuscript.

3, In the categorization of PLN or MLN, authors described LN class III, IV, and V. Is this classification based on the ISN/RPS 2003, correct? Please mention the famous classification for LN in the manuscript. In addition, do authors include the patients with combined class III/V + V as PLN? If not, please clarify that LN patients in combined class III/IV +V were not contained in the subjects of PLN.  

We added some clarification on this and included references to both ISN/RPS and WHO classification in the Introduction. The subjects were classified by WHO classification, but patients with Class V disease were only included if “pure class V” and we did not include class V LN with proliferative lesions.  

4, Authors need to simply explain the method of statistical analysis in the section of material method.

A Statistical Analysis Method section 2.7 has been included.

5, Please show the unit of the graph in Figure 3c. Authors mentioned as follow; Percent of immunostaining in each glomerular tuft was determined by the ratio of immunostained area to total glomerular tuft surface area. I think that staining area in representative photo in Figure 3a is over 25%?   

The level of staining to be captured was set and compared to the total glomerular area.  The level of detection was set to limit non-specific background staining and was utilized for all glomeruli in all sections. There was a range in staining for each individual subject (see figure 3d) and each glomerulus is represented by a single dot. There were several glomeruli that were around 15-20% in the PLN samples.

Reviewer 2 Report

In this manuscript, the authors showed that moesin expressed in glomeruli is recognized by antibodies from patients suffering with lupus nephritis. 

Auto-antibodies in lupus are found in several Ig classes. The authors focused on IgG. What about IgM and IgE against moesin?

The authors should provide a table with clinical characteristics of patients. Are anti-moesin antibody levels associated with proteinuria? Or moesin staining with proteinuria? 

Some controls are missing in results section. Figure 1: can the authors provide a blot with NC? Figure 3, can the authors provide staining of moesin on LC sample? The number of patients is not mentioned in figure legends.

In materials and methods section, there is no description of statistical analyses.

Author Response

Reviewer 2

In this manuscript, the authors showed that moesin expressed in glomeruli is recognized by antibodies from patients suffering with lupus nephritis. 

Auto-antibodies in lupus are found in several Ig classes. The authors focused on IgG. What about IgM and IgE against moesin?

We chose to focus on IgG as IgG is the most abundant antibody in human serum.  We agree it would be interesting to evaluate other antibody classes in the future, but it was outside the scope of this study.

The authors should provide a table with clinical characteristics of patients. Are anti-moesin antibody levels associated with proteinuria? Or moesin staining with proteinuria? 

A table with clinical characteristics has been added with clinical data when available. All PLN subjects have clinical data and were in flare at time of serum collection.

Some controls are missing in results section. Figure 1: can the authors provide a blot with NC? Figure 3, can the authors provide staining of moesin on LC sample? The number of patients is not mentioned in figure legends. 

A blot from normal control has been added. We have increased the number of samples used for IHC in figure 3 and added MLN (class V) samples. We are unable to add lupus control samples as we do not have kidney biopsy specimens from patients with SLE and without nephritis. The number of patients has been added to figure legends.

In materials and methods section, there is no description of statistical analyses. 

A Statistical Analysis Method section is now included.

Round 2

Reviewer 1 Report

As a result of careful consideration, revised manuscript by authors improved dramatically although several points remained still unconvincing. However, such unclarified points were described as future issues to be uncovered or important limitation. Taken together, revised manuscript reached to the level to be accepted. I did not demand additional questions or suggestions.

Reviewer 2 Report

The authors answered to all the point.

This manuscript is a resubmission of an earlier submission. The following is a list of the peer review reports and author responses from that submission.